# Generation and Absorption of Periodic Waves Traveling on a Uniform Current in a Fully Nonlinear BEM-based Numerical Wave Tank

**Dimitris I. Manolas [1,2,*], Vasilis A. Riziotis [1] and Spyros G. Voutsinas [1]**

[1] School of Mechanical Engineering, National Technical University of Athens,
GR15780 Zografos, Athens, Greece; vasilis@fluid.mech.ntua.gr (V.A.R.); spyros@fluid.mech.ntua.gr (S.G.V.)
[2] iWind Renewables PC, GR15344 Gerakas, Athens, Greece
[*] Correspondence: manolasd@fluid.mech.ntua.gr; Tel.: +30-210-772-1097

**Abstract:** Accurate and efficient numerical wave generation and absorption of two-dimensional nonlinear periodic waves traveling on a steady, uniform current were carried out in a potential, fully nonlinear numerical wave tank. The solver is based on the Boundary Element Method (BEM) with linear singularity distributions and plane elements and on the mixed Eulerian–Lagrangian formulation of the free surface equations. Wave generation is implemented along the inflow boundary by imposing the stream function wave solution, while wave absorption at both end-boundaries is effectively treated by introducing absorbing layers. On the absorbing beach side, the outflow boundary condition is modified to ensure that the solution accurately satisfies the dispersion relation of the generated waves. The modification involves a free-parameter that depends on the mass flux through the domain and is determined through a feedback error-correction loop. The developed method provides accurate time domain wave solutions for shallow, intermediate, and deep water depths of high wave steepness (wave heights up to 80% of the maximum value) that remain stable for 150 wave periods. This also holds in case a coplanar or opposing uniform current of velocity up to 20% of the wave celerity interacts with the wave.

**Keywords:** fully nonlinear numerical wave tank; wave generation and absorption; periodic waves; boundary element method; wave–current interaction; stream function; absorbing layers

## 1. Introduction

Over the years, the design of offshore structures, such as platforms and more recently wind and wave energy plants, has been mainly based on linear hydrodynamic theory. Random waves are usually considered as a Gaussian process, while the response of the structure is assumed to be linear. By using second-order theory, the validity range of a linearized approach is extended; however, accounting for higher order nonlinear waves still requires the use of safety factors that may oversize the designs. In order to reduce the uncertainty of the design and reduce safety factors and cost, more advanced numerical models should be formulated. Consequently, over the last decades, significant progress has been made in the direction of the accurate and efficient solution of the fully nonlinear water wave problem [1]. Numerical tools have been developed, satisfying the dynamic and kinematic nonlinear boundary conditions at the instantaneous position of the free surface, while considering accurate numerical time integration schemes capable of providing accurate solution of the nonlinear water wave problem. The aim of the so-called numerical wave tanks (NWT) is to reproduce the experimental results, using numerical simulations. The developed methods accurately simulate the wave generation and absorption, as well as the dynamic response and wave loading of fixed or floating structures.

Models based on the assumption of the inviscid and irrotational flow, that however retain the nonlinear character of the problem, have been proven to be particularly successful in simulating nonlinear water waves and their interaction with structures, and at the same time computationally less demanding as compared to Navier–Stokes solvers. Therefore, potential solvers are still attractive being a fair trade-off between accuracy and computational cost and are widely used by the industry. Detailed reviews on fully nonlinear potential solvers can be found in References [2–5], most of which are connected to the Boundary Element Method (BEM), as well as in the more recent Reference [6].

The first pioneering work in numerical nonlinear wave hydrodynamics was the development of the mixed Eulerian–Lagrangian formulation by Longuet-Higgins and Cokelet [7]. In this approach, the two nonlinear free surface boundary conditions were integrated in time, to estimate the evolution of the free surface in the Lagrangian frame, while the unknown normal velocity of the Lagrangian markers of the free surface was calculated in the Eulerian frame by solving the Laplace equation. In their work, waves were generated by applying a pressure distribution on the free surface, which was introduced in the dynamic boundary condition, while periodic conditions were imposed to the two lateral boundaries, preventing the onset of artificial reflections.

In case periodic conditions are not applicable (e.g., because of the presence of a submerged obstacle), the conditions at the lateral end-boundaries should be non-reflective. In early developments, the Sommerfeld/Orlanski boundary condition was employed [8], which exhibited numerical instabilities especially in transient flows due to irregular waves or due to the presence of flow disturbances, as indicated by Romate [9]. Consequently, alternative wave-absorption techniques were developed basically inspired by those used in wave tank facilities, either by modeling the physical wave absorber or by introducing the absorbing effects in the free surface equations through additional damping terms. Along the first option, Bessho [10] and Naito [11] modeled a plunger-type physical wave absorber, and Clément [12] considered a piston-type physical absorber suitable for long waves. Along the second option, Baker et al. [13] and Cointe et al. [14] followed the idea of Israeli and Orszag [15] and Le Méhauté [16] and introduced dissipative terms in the dynamic and the kinematic free surface boundary conditions. Various versions exist that differ in the choice of the damping function, the considered variable (i.e., the potential or its normal derivative), and the equation in which dissipation is added (i.e., in both free surface equations or only in one of them). A combination of the two methods was proposed by Clément [12], who considered a piston absorber in conjunction with an absorbing layer. Clamond et al. [17] considered a modified dynamic boundary condition at the free surface to damp the tangential velocity, in combination with a damping term in the kinematic boundary condition. They efficiently damped a steep solitary wave, as well as waves due to advancing pressure distributions in three dimensions. Recently, Spinneken, Christou, and Swan [18] considered an active flap absorber based on a force-feedback control loop in a fully nonlinear BEM-based NWT. They employed the multiple-flux technique [19] and simulated the absorption of regular waves, focused wave groups, and random waves.

In NWTs, waves are often generated by replicating the motion of the actual wave maker over the inflow boundary either as a flap or a piston wave maker [3,5,14]. The motion is usually estimated based on first- or second-order solutions of the so-called wave-maker problem (see Reference [5] for example). Therefore, as the wave height increases and exceeds the range of validity of the adopted theory, resonant nonlinear interactions create higher-order harmonic modulation of the wave shape that render impossible the generation and propagation of steady progressive waves without modulation of their shape, as it is also observed in water-tank tests [20–22].

A way to overcome the above shortcoming is to impose a valid wave solution to the inflow boundary of the NWT. Airy or Stokes second-order wave kinematics have been widely considered due to the compact representation of the solution they provide in closed form. Because both theories are based on expansions and retain terms up to a certain order, neither the Airy nor Stokes theory is accurate for high wave steepness, and modulation of the wave shape will again appear. This is

reported by Ryu, Kim, and Lynett [23], who studied the wave–current interaction in a fully nonlinear NWT by generating the incoming wave based on Airy theory.

On the other hand, by imposing a stream function (SF) nonlinear wave solution [24–26] to the inflow boundary of the NWT, (a) a numerically exact wave solution is imposed that is valid over the whole water depth and wave height range, (b) there is no need to apply any stretching/extrapolation method to the imposed solution since it is inherently valid up to the instantaneous position of the free surface, and (c) the nonlinear wave–current interaction problem in the case of a steady, uniform current is consistently accounted for. Moreover, solutions provided by SF theory and fully nonlinear NWT solver are compatible, because both methods solve the complete water wave problem (by satisfying the nonlinear free surface boundary conditions on the instantaneous position of the free surface, under the assumption of inviscid and irrotational flow). Thus, the generated waves can propagate in the NWT computational domain without modulation of their shape, as long as the numerical errors due to discretization of the boundary (in BEM-based NWT) and truncation of the Fourier series (in SF theory) are small.

In nonlinear wave theory, the mean wave mass transport is non-zero, as opposed to the Airy theory. Fluid mass is transported in the direction of wave propagation, called Stokes' drift. Consequently, extra mass flux crosses the inflow boundary over a wave period that gradually increases the fluid mass contained in the computational domain depending on the outlet conditions that are applied. Unless the extra mass flux is properly accounted for in the NWT, the numerical simulations become inaccurate or blow up due to the violation of the mass conservation principle. A simple way to overcome the above difficulty was originally proposed by Grilli and Horrillo [27]. It requires zero net mass flux over the wave period as an extra constraint to the formulation of the SF method, which defines an opposing uniform current. This cancels out Stokes' drift and can be achieved by setting equal to zero the vertically integrated mean transport velocity $U_S$ over the wave period [26]. A more general approach, which is proposed in the present paper, consists of properly modifying the boundary condition on the end side wall (absorbing beach side) so that mass flux may pass through the computational domain. This permits to accurately generate and propagate nonlinear periodic waves that are not necessarily restricted by the zero mass flux requirement.

Klopman [28] was the first to consider SF waves in a fully nonlinear BEM-based solver, while Ferrant [29] estimated the wave run-up on a cylinder due to waves and current in a fully nonlinear NTW in three dimensions. In his work, the incoming flow was specified based on the SF theory by Rienecker and Fenton [25]. Grilli and Horrillo [27] performed an in-depth study of the numerical wave generation and absorption of periodic waves based on the SF solution. They used a fully nonlinear NWT based on higher-order BEM and generated zero mass flux SF waves. In their implementation, the inflow vertical boundary was moved in the horizontal direction through an iterative procedure, in order to deal with the poor resolution of the surface panels near the inflow boundary due to drift phenomena. They considered damping terms only in the free surface dynamic equation and only at the absorbing beach side, proportional to $\partial_n\varphi$, with a time-varying damping coefficient to optimize absorbing efficiency. They reported reduced absorbing performance for low-frequency (long) waves, and they encountered reflections from the end-boundary. In order to overcome the abovementioned reflections, they considered an active piston absorber, in combination with the absorbing layer. They generated nonlinear waves with wave heights corresponding to 52% and 42% of the maximum value for non-dimensional wavelengths ($\lambda/d$) equal to 2.1 and 10.3, respectively.

The present work investigates the numerical and modeling aspects appearing in potential NWTs and aims at (a) accurate and stable numerical generation of periodic waves with very high wave steepness at a wide water depth regime, (b) modeling of the nonlinear wave–current interaction, and (c) effective wave absorption at the open boundary on the beach side, without affecting the nonlinear dispersion relation of the generated waves. In this regard, the wave generation, propagation, and absorption of periodic waves with very high wave steepness, interacting with a steady uniform current, are considered in a fully nonlinear potential BEM-based NWT, using piecewise linear singularity

distributions and plane elements. The contribution of the current is consistently considered within the SF theory, permitting the generation in the NWT of periodic waves traveling on a steady, uniform current that are not restricted by the zero mass flux requirement. The horizontal position of the two vertical boundaries of the computational domain is held fixed, by moving the surface nodes at both ends only vertically, based on the semi-Lagrangian approach, while all other free surface nodes are tracked, following the Lagrangian approach. Regridding of the surface nodes is performed, based on spline fitting, in order to maintain an almost constant panel resolution and to increase the accuracy of the BEM computation. Wave reflection at the two ends of the domain is efficiently prevented by introducing damping terms in both free surface equations that dissipate wave energy over absorbing layers of finite width. Additionally, in order to achieve an accurate wave characterization (period and wavelength), the condition on the vertical end-boundary is modified to include an unknown mass flux. This additional free parameter is estimated by using a feedback correction loop based on Proportional Integral (PI) control.

The developed method is successfully tested in wave generation at shallow, intermediate, and deep water depths ($\lambda/d = 1, 7, 13$) for wave heights that correspond to 80% of the maximum value (see Equation (25) defined in Reference [30] for the definition of the maximum wave height per water depth), as well as in the case the wave travels on an opposing or a coplanar steady, uniform current of non-dimensional velocity $U_0/c = \pm 0.2$. In all cases, wave absorption is very efficient, while the simulations remain stable for at least 150 wave periods (time instant at which the numerical simulation is forced to stop by the user). The novelty of the present contribution lies in the consistent and stable modeling of high periodic waves interacting with a steady, uniform current in a fully nonlinear potential NWT. This was accomplished by thoroughly working out the numerical implementation details. In particular, the modified condition at the end-boundary allows nonlinear, periodic wave solutions to be accurately represented in the NWT, without being affected by the presence of the absorbing layer.

## 2. Mathematical Formulation

The potential irrotational flow of an incompressible fluid without surface tension is considered in the physical flow domain, $D(t)$, with boundaries collectively denoted as $S(t)$ (Figure 1). The 2D problem is considered in the vertical plane $(x, z)$, where $z = 0$ is the mean water level. $S_{\mathrm{FS}}(t)$ is the free surface boundary, expressed as $z = \zeta(x; t)$; $S_{\mathrm{SB}}$ is the seabed, while $S_{\mathrm{WG}}(t)$ and $S_{\mathrm{AB}}(t)$ are the vertical boundaries on the wave generator and the absorbing beach side, respectively.

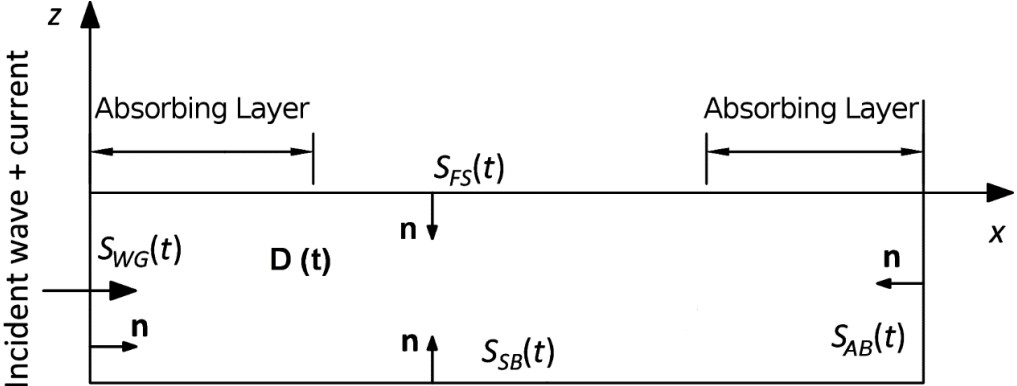

**Figure 1.** Definition of the computational domain and the boundary surfaces.

The total scalar potential, $\varphi(\mathbf{x}; t)$, is defined as follows:

$$\varphi(\mathbf{x}; t) = \varphi_w(\mathbf{x}; t) + U_0 x \tag{1}$$

where $\varphi_w(\mathbf{x};t)$ denotes the wave potential, and $U_0$ denotes the velocity of the steady, uniform current.

The total potential satisfies the following set of equations (see eg References [1–5,7]):

$$\nabla^2\varphi = 0, \; in \; D(t) \tag{2}$$

$$\frac{\partial\varphi}{\partial t} = -g\,\zeta - \frac{1}{2}|\nabla\varphi|^2 - \frac{p_a}{\rho}, \; on \; S_{FS}(t) \tag{3}$$

$$\frac{\partial\zeta}{\partial t} = -\nabla\varphi\cdot\nabla\zeta + \frac{\partial\varphi}{\partial z}, \; on \; S_{FS}(t) \tag{4}$$

$$\frac{\partial\varphi}{\partial n} = 0, \; on \; S_{SB} \tag{5}$$

$$\frac{\partial\varphi}{\partial n} = U_0\,n_x, \; on \; S_{AB}(t) \tag{6}$$

$$\frac{\partial\varphi}{\partial n} = u_x^{SF}(\mathbf{x}_G;t), \; on \; S_{WG}(t) \tag{7}$$

Laplace Equation (2) represents mass conservation in the computational domain, while the Bernoulli Equation (3), also known as the free surface dynamic boundary condition, assures pressure continuity across $S_{FS}(t)$. The kinematic boundary condition (4) states that the fluid particles on the free surface always move along the free surface. Both conditions (3) and (4) are applied on the instantaneous free surface position at $z = \zeta(x;t)$. On the seabed $S_{SB}$ and on the absorbing beach $S_{AB}(t)$, the no-penetration condition is satisfied through Equations (5) and (6). No-penetration condition on the absorbing beach is only applied on the wave velocity potential, to allow the current to pass through. Along the inflow boundary, $S_{WG}(t)$, the wave and the current are introduced though Equation (7), by matching the known normal velocity, $u_x^{SF}$, as defined by the stream function solution. Both vertical boundaries $S_{AB}(t)$ and $S_{WG}(t)$ are placed at fixed x-positions.

In the abovementioned set of equations, $g$ denotes the acceleration due to gravity, $p_a$ the atmospheric pressure (which is set equal to zero), $\rho$ the fluid density in $D(t)$, $\mathbf{n}$ the normal to the surface boundary unit vector always pointing toward the flow domain, and $n_x$ the horizontal component of $\mathbf{n}$. The time-varying integration constant appearing in the Bernoulli equation was eliminated, leading to a proper redefinition of $\varphi(\mathbf{x};t)$.

## 3. Numerical Implementation

### 3.1. Mixed Eulerian Lagrangian Method

Free surface boundary conditions (3) and (4) are written in Eulerian form, while they are equivalently expressed in Lagrangian form (see eg References [1–5,7]),

$$\frac{d\varphi}{dt} = -g\,\zeta + \frac{1}{2}|\nabla\varphi|^2 - \frac{p_a}{\rho} - damp_1 \tag{8}$$

$$\frac{d\mathbf{x}}{dt} = \nabla\varphi - damp_2\,\delta_{iz}, \tag{9}$$

or in 'semi-Lagrangian' form,

$$\frac{d\varphi}{dt} = -g\,\zeta - \frac{1}{2}|\nabla\varphi|^2 + \frac{d\zeta}{dt}\frac{\partial\varphi}{\partial z} - \frac{p_a}{\rho} - damp_1 \tag{10}$$

$$\frac{d\zeta}{dt} = -\nabla\varphi\cdot\nabla\zeta + \frac{\partial\varphi}{\partial z} - damp_2, \tag{11}$$

where artificial damping terms $damp_1$ and $damp_2$ were added (they are defined in Section 3.5). In the above equations, $\mathbf{x}$ denotes the position vector of the markers on the free surface $\mathbf{x} = (x, \zeta)$, $\delta_{ij}$

denotes the delta of Kronecker, and $d(.)/dt = \partial(.)/\partial t + \mathbf{Up} \cdot \nabla(.)$ denotes the material derivative. In the Lagrangian formulation, the markers on the free surface are tracked as material particles, and, therefore, their velocity, $\mathbf{U_p}$, is equal to the total velocity, $\nabla\varphi$ (which includes the current term). In the semi-Lagrangian formulation, the markers are only allowed to move in the vertical direction, and, therefore, $\mathbf{U_p}$ is equal to $(0, 0, d\zeta/dt)$. In either formulation, the two end-points of the free surface are tracked, using the semi-Lagrangian formulation, in order to maintain their horizontal position.

The full and semi-Lagrangian formulations are equivalent provided that the surface elevation is a single valued function, while only the Lagrangian formulation can simulate wave breaking until the time instant at which the overturning crest reattaches to the free surface. Moreover, the derived expressions of the two free surface boundary conditions are simpler in the Lagrangian formulation, in which $|\nabla\varphi|^2$ is the only nonlinear term. In the semi-Lagrangian formulation, two additional nonlinear terms appear (i.e., terms $\nabla\varphi \cdot \nabla\zeta$ and $\frac{d\zeta}{dt}\frac{\partial\varphi}{\partial z}$). In this regard, the Lagrangian formulation may increase numerical stability, especially for high values of the wave steepness, that increases the significance of the nonlinear terms. However, the Lagrangian formulation requires regridding of the nodes on the free surface, as described in Section 3.4.

For given $\mathbf{x}$ or $\zeta$ (depending on the formulation) and $\varphi$ over the free surface, the above evolution equations are integrated in time by means of a Runge–Kutta 4th-order explicit scheme, providing new position and free surface potential. In either formulation, the Laplace equation acts as constraint, while its solution can be conveniently introduced in convolution form, as detailed in Section 3.3.

### 3.2. Initial Conditions and Ramp Function

At $t = 0$, either calm conditions ($\varphi = 0$ and $\zeta = 0$) are set on the free surface or a valid solution is imposed, in order to save computational time. In the former case, a ramp function, $F_{ramp}(t)$ [1–5],

$$F_{ramp}(t) = \tanh(\mu t), \quad \mu = \frac{2.5}{T_{ramp}} \tag{12}$$

is introduced, in order to avoid excessive transient accelerations that could cause numerical instabilities. $F_{ramp}(t)$ varies from 0 to 1 and usually spans over 2 to 3 wave periods $T_{ramp}$. $F_{ramp}(t)$ gradually increases the applied inflow wave and current field (terms $u_x^{SF}$, $\partial_n\varphi^{SF}$, $\zeta^{SF}$ in Equations (7), (16), and (17)).

### 3.3. Integral form of the Laplace Equation and Its Numerical Solution

The solution of the Laplace Equation assumes the following representation [31]:

$$a(\mathbf{x_0})\,\varphi(\mathbf{x_0}) = \int_S \left[ \frac{\partial\varphi(\mathbf{x})}{\partial n}G(\mathbf{x};\mathbf{x_0}) - \varphi_S(\mathbf{x})\frac{\partial G(\mathbf{x};\mathbf{x_0})}{\partial n} \right] dS(\mathbf{x}) \tag{13}$$

$$G(\mathbf{x};\mathbf{x_0}) = \frac{1}{2\pi}\ln|\mathbf{x} - \mathbf{x_0}| \tag{14}$$

where $\varphi_S$ is used in the integral to denote the restriction of $\varphi$ on $S$, in order to designate that this distribution only admits surface gradients. $G(\mathbf{x}, \mathbf{x_0})$ is the Green function, $\mathbf{x_0}$ denotes any field point in $D(t) \cup S(t)$, $\mathbf{x}$ any point on $S(t)$, and $a(\mathbf{x_0})$ the solid angle at $\mathbf{x_0}$. For points in $D(t)$, $a(\mathbf{x_0})$ equals to $2\pi$, while, for points on $S(t)$, $a(\mathbf{x_0})$ is directly calculated by setting, in Equation (13), the following uniform boundary conditions: $\partial_n\varphi = 0$ and $\varphi_S = 1$ [31]. When applied on $S(t)$, the integral equation only involves boundary values of the potential and its normal derivative; therefore, depending on the type of the boundary condition (Dirichlet or Neumann), this integral Equation associates $\varphi_S$ to $\partial_n\varphi$ or vice versa on every part of $S(t)$. Dirichlet conditions are imposed along the free surface, while Neumann conditions are applied on the other boundaries. The resulting mixed problem is numerically solved by using BEM, by applying piecewise linear approximations for $\varphi_S$ and $\partial_n\varphi$. The collocation points

are placed at the nodes of the surface grid. By assuming plane panels, integrals in Equation (13) are derived analytically [32], while its discrete form is solved, using the LU decomposition method.

### 3.4. Spatial Derivatives, Double Nodes Representation, and Regridding of Boundaries

Spatial derivatives $\partial_s\varphi$ and $\partial_x\zeta$ on $S_{FS}(t)$ that appear in Equations (8)–(11) are numerically obtained at the boundary surface grid nodes by means of either spline interpolations on the $\varphi$ and $\zeta$ values of the grid points (degrees of freedom of numerical solution) and analytical differentiation thereafter or by directly applying finite differences (e.g., 2nd-order central finite difference approximations for non-uniform grid spacing) [1–5].

Special treatment is required at the end-nodes of the free surface, where the boundary condition switches from Dirichlet to Neumann (i.e., at the intersection of the free surface and the vertical end-walls). In order to uniquely define the velocity and prevent the onset of saw-tooth instabilities, these particular nodes are considered as "double nodes" [3,33]. At the double nodes, the velocity is expressed in terms of the two known normal components, $\partial_n\varphi^D$ and $\partial_n\varphi^N$, defined at the end-points of the Dirichlet and the Neumann boundary sides, respectively. At the Dirichlet boundary (free surface), $\partial_n\varphi^D$ is obtained through the solution of the boundary integral equation (BIE) (13), while at the Neumann boundary (side boundaries), $\partial_n\varphi^N$ is assumed known from the boundary data. Based on this information, the velocity at the double node $\mathbf{u} = (u_x, u_z)$ is obtained by solving the following $2 \times 2$ linear system:

$$\begin{aligned}
\mathbf{u} \cdot \mathbf{n}^D &= u_x\, n_x^D + u_z\, n_z^D = \partial_n\varphi^D \\
\mathbf{u} \cdot \mathbf{n}^N &= u_x\, n_x^N + u_z\, n_z^N = \partial_n\varphi^N
\end{aligned} \tag{15}$$

Another thing of importance is the regridding of the free surface nodes in order to retain an almost uniform spacing, when solving the integral Equation (13) in the Lagrangian context [1–5]. It is based on spline fitting and is performed when the ratio between the maximum and the minimum panel length exceeds certain preselected limits, usually set in the range of (1.2, 1.5). Moreover, for numerical consistency and accuracy, regridding of both vertical boundaries is performed in every time step, based on the instantaneous surface elevation, in order to ensure that $S(t)$ remains a closed surface. In this way, artificial smoothening, as considered for example in Reference [7], based on the Chebychev 5- or 7-point scheme, is not needed.

### 3.5. Absorbing Layers

In order to avoid reflections at the end-boundaries of the flow domain, wave absorption is applied. On the beach, side an absorbing layer extending to 2 wavelengths is defined. Within this layer, gradually increasing damping terms are introduced in both free surface boundary conditions. An absorbing layer extending to 1 wavelength is also defined at the inflow side, to prevent wave breaking and thus to provide numerical stability in highly nonlinear cases that approach the wave-breaking limit.

On the wave-generator side, the *damp*$_i$ terms added in free surface Equations (8)–(11) are defined as follows (see eg References [1–5]):

$$damp_1 = v_1(x)(\partial_n\varphi - \partial_n\varphi^{SF}) \tag{16}$$

$$damp_2 = v_2(x)(\zeta - \zeta^{SF}) \tag{17}$$

where $\partial_n\varphi^{SF}$ and $\zeta^{SF}$ correspond to the stream function expressions of the incident wave. They are introduced in order to only dissipate the reflected waves instead of the total wave. On the beach side, the total wave should be absorbed, so $\partial_n\varphi^{SF}$ and $\zeta^{SF}$ are set equal to zero. The $v_i(x)$ coefficient is a function of space and has the following form [14]:

$$v_i(\mathrm{x}) = \alpha_i\omega\left(\frac{|x - x_e|}{L_d}\right)^{b_i} \tag{18}$$

The absolute value ensures positive $v_i(x)$ on both sides, while exponents higher than 1 give an increasing damping effect as the outer boundaries are approached. $\omega$ is the wave frequency, $L_d$ is the length of the absorbing layer, and $x_e$ defines the position where the absorbing layer ends or starts, while $\alpha_i$ and $b_i$ are tuning coefficients defined as follows: $\alpha_1 = (1/2\,k_0)$, $\alpha_2 = 1$, $b_1 = b_2 = 2$. $k_0$ is the local wave number at the absorbing layer depth.

### 3.6. Modification of the Outflow Condition on the Absorbing Beach Boundary

The numerically generated nonlinear, periodic waves will slightly deviate in wavelength, as compared to the standard periodic formulation (i.e., without absorbing layers), and the nonlinear dispersion relation will not be accurately satisfied in case condition (6) is applied on the absorbing beach boundary, $S_{AB}(t)$, which is considered alongside with the absorbing layer. Although dispersion errors may be triggered by space and time resolution, such an effect can be minimized by proper grid and time-step sensitivity study. This means that, ultimately, dispersion errors are attributed to the effect the absorbing layers have on the solution, as compared to the standard periodic formulation, as detailed in the numerical results of Section 4. Thus, in the present work, boundary condition (6) is modified to allow additional mass flow (not only due to the steady current) passing through $S_{AB}(t)$:

$$\frac{\partial \varphi}{\partial n} = \left[ U_0 + c_{flux}(U_s - U_0) \right] n_x, \text{ on } S_{AB} \tag{19}$$

where $n_x$ is the horizontal component of the normal unit vector and is equal to $-1$ on $S_{AB}(t)$ (see Figure 1), and $c_{flux}$ is the coefficient that determines the outgoing/incoming additional net mass flux through the outflow boundary on the beach side. If $c_{flux} = 0$, (19) reduces to (6), and if $c_{flux} = 1$, there is additional mean mass outflow (due to Stokes' Drift), while if $c_{flux} < 0$ net mass enters to domain $D(t)$ through $S_{AB}$. $U_S$ is the mean mass transport velocity integrated over the wave period, $T$, and over the local water depth, $d_0$, at the horizontal position, $x_0$, defined according to References [27,34] as follows:

$$U_s = \frac{1}{T\,d_0} \int_0^T \int_{-d_0}^{\zeta} \partial_x \varphi(x_0, z, t)\, dz\, dt \tag{20}$$

where $U_s$ also includes the current contribution as it appears in the definition of the total velocity potential in Equation (1). In the present work, $U_S$ is estimated based on SF theory, while it can also be estimated in BEM by integrating the flux along the inflow boundary, $S_{WG}$.

In order to automatically correct the flux coefficient, a feedback control loop is added, using a linear PI control element. The input signal is the phase error of the surface elevation between the SF and the NWT solutions, while the controlled signal is the flux coefficient in Equation (19). The tphase error is calculated through the Fourier series of the surface elevation time series at a selected point before the absorbing beach (i.e., at the point where the absorbing beach starts) over the last wave period. The block diagram of the control loop is shown in Figure 2.

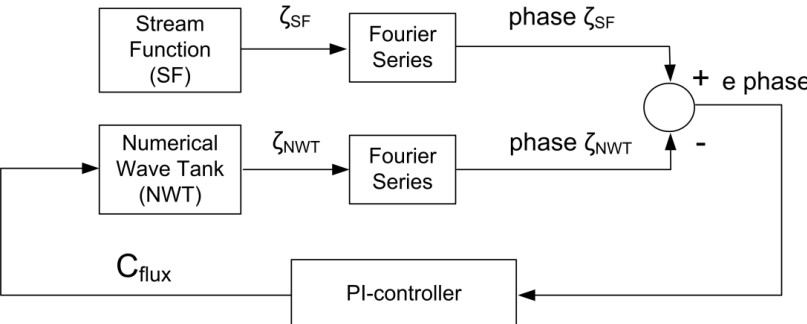

**Figure 2.** Block diagram of the closed control loop for correcting the flux coefficient.

### 3.7. Wave Kinematics According to Stream Function Theory

The SF implementation used in the present solver is based on Fenton's work [26]. The 2D incompressible irrotational flow is considered in a frame moving with the wave celerity, $c$. In the above moving frame, the problem is steady and only solved once. The wave is assumed symmetric and periodic, traveling on a depth-uniform current $U_0$. In this regard, the nonlinear dependence of the wave characteristics on the current velocity is inherently accounted for, as discussed in Section 4.3. An SF $\psi$ exists that satisfies the Laplace equation, the bottom kinematic boundary condition, and the free surface dynamic and kinematic boundary conditions, both imposed at the exact instantaneous free surface position, as well as periodic conditions in the lateral boundaries. The core of the method is the expansion of $\psi$ in Fourier series, while the only approximation is the truncation of the infinite series after $N$ terms. The two nonlinear free surface equations are evaluated at $N + 1$ equispaced points over half wavelength due to symmetry, while the nonlinear system of equations is solved by using Newton's method. The surface elevation and the horizontal and vertical wave velocities are expressed in the physical frame as follows:

$$\zeta^{SF}(x_G;t) \;=\; k^{-1}\sum_{j=1}^{N}{}'' \left[ \frac{2}{N}\sum_{m=0}^{N}{}'' k\eta_m \cos\!\left(\frac{jm\pi}{N}\right) \right] \cos\!\left(jk(x_G - ct)\right) \tag{21}$$

$$u_x^{SF}(\mathbf{x}_G;t) \;=\; U_0 + \sqrt{\frac{g}{k}}\sum_{j=1}^{N} jB_j \frac{\cosh jk(z_G+d)}{\cosh jkd}\cos jk(x_G - ct) \tag{22}$$

$$u_z^{SF}(\mathbf{x}_G;t) \;=\; \sqrt{\frac{g}{k}}\sum_{j=1}^{N} jB_j \frac{\sinh jk(z_G+d)}{\cosh jkd}\sin jk(x_G - ct) \tag{23}$$

In the above equations, $k\eta_m$ represents the non-dimensional unknown surface elevations, and $B_j$ represents the unknown Fourier coefficients, both estimated within the solution procedure. The term in square brackets in Equation (21) is the cosine transform of the $N + 1$ surface elevations, while $\Sigma''$ denotes the trapezoidal-type summation with factors of $\frac{1}{2}$ multiplying the 0th and the Nth contributions.

### 3.8. Description of the Solver

The core of the solver is the stepwise integration in time of the two nonlinear free surface boundary conditions (see Equations (8)–(11) depending on the formulation), which, based on an explicit (Runge–Kutta) scheme, can be written in compact form:

$$\mathbf{u}^{t+dt} \;=\; \mathbf{u}^t + \frac{d\mathbf{u}}{dt}^t dt \tag{24}$$

where $\mathbf{u} = \{\varphi_{FS}, \mathbf{x}_{FS}\}^{\mathrm{T}}$ is the vector of the unknowns containing the potential and the position vector of every marker on the free surface (in semi-Lagrangian formulation $\mathbf{x}_{FS}$ reduces to $\zeta$). In the above explicit scheme, the left-hand side refers to the next time instant, $t + dt$, while the right-hand side to the current time instant, $t$. To advance in time, the unknown material derivatives need to be calculated at time, $t$, assuming that the boundary surface $S(t)$, the Dirichlet data on $S_{FS}(t)$, and the Neumann data along the other boundaries are known.

The main tasks performed in every time instant, $t$ (i.e., in every sub-step $t_i$ of the Runge–Kutta method), are listed in the sequel. They are divided into three parts; the definition/generation of the boundary surface, the solution of the BIEs, and the stepwise integration in time of the free surface boundary conditions.

1. Definition/Generation of Boundary Surface $S(t)$

   a. Regridding of the nodes on $S_{FS}(t)$ (if needed) and of the vertical boundaries $S_{WG}(t)$, $S_{AB}(t)$ (every time step) based on the instantaneous surface elevation at the end-nodes;
   b. Generation of $S(t)$ to be used in BEM.

2. Solution of BIEs Using BEM (Eulerian Part)

   a. Calculation of the convolution integrals in BIE (13) and of the solid angle, $a$;
   b. Assignment of the known (Dirichlet or Neumann) data on $S(t)$, i.e., on $S_{WG}(t)$ from the stream function theory and on $S_{FS}(t)$ from the previous time step solution;
   c. Formulation and solution of the linear system of equations to estimate $\partial_n\varphi$ on $S_{FS}(t)$;
   d. Calculation (numerically) of $\partial_s\varphi$ and $\partial_x\zeta$ ($\nabla\zeta$) on $S_{FS}(t)$;
   e. Calculation of $\nabla\varphi$ on $S_{FS}(t)$ (transformation from the local coordinate system $(s,t)$ to the global one $(x,z)$ and application of the double-node technique at the end-nodes).

3. Integration in Time (Lagrangian Part)

   a. Calculation of the total derivatives of $u(t)$, based on the estimated $\nabla\varphi$ and $\nabla\zeta$ on $S_{FS}(t)$, considering the damping terms that appear within the absorbing layers;
   b. Estimation of the potential and position vector of the markers on $S_{FS}(t)$ at $t + dt$.

## 4. Numerical Results and Discussion

### 4.1. Validation of the Method Against Measurements

The interaction of periodic waves with variable bathymetry, defined through a submerged trapezoid bar, is considered in order to validate the numerical method and its implementation. Figure 3 shows the computational domain that corresponds to the tunnel tests conducted by Beji and Battjes [35] (depth 0.4 m and length 30 m). The length of the absorbing layer (damping zone) is set to 4 and 8 m at the wave generator and the absorbing beach side, respectively. A stream function periodic wave solution is matched along the inflow boundary with wave height H = 1.905 cm and period T = 2.02 s. A grid independence study has led to 1800 nodes along the free surface, 650 at the seabed, and 30 at the side boundaries, in combination with 200 time steps per wave period (dt = 0.0101 s). This dense discretization is used in order to capture the generated higher harmonics. The ramp function is defined over two periods.

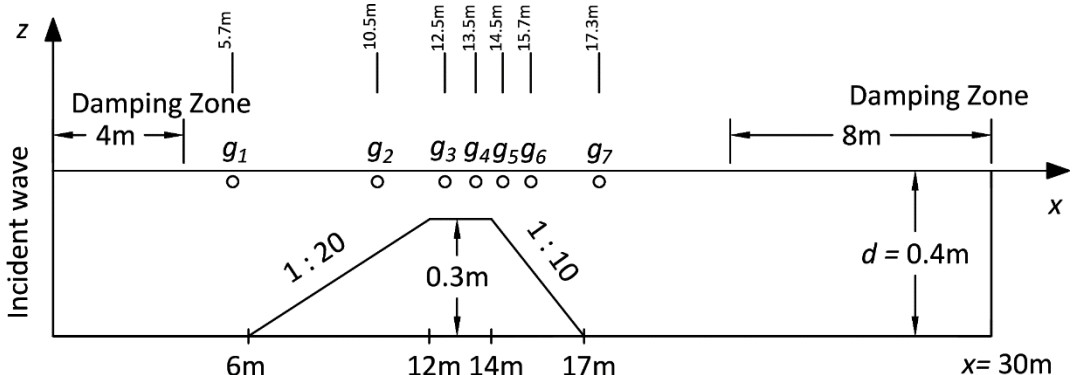

**Figure 3.** Definition of the Boundary Element Method (BEM) computational domain in case of periodic waves interacting with a submerged bar. Points g1 to g7 denote the position of the wave gauges.

In Figure 4, the time series of the free surface elevation are compared against the experimental data at specific stations (wave gauges 1 to 7). A linear solution is also plotted in order to demonstrate

the importance of nonlinearities. It is derived by considering the geometry of the free surface fixed, by linearizing the two free surface boundary conditions (Equations (3) and (4)), and by applying wave kinematics based on Airy theory along the inflow boundary. Due to shoaling, energy is transferred to higher harmonics in the shoaling region, which, however, remains bounded. These higher harmonics are released in the deepening part after the bar, creating an irregular pattern. The agreement between the predictions of the nonlinear NWT and the experiment is very good in all wave gauges, while linear theory clearly fails to reproduce the present phenomenon. The higher frequencies that appear initially at the first wave gauge are the main source of minor differences at the following stations. The present test verifies that the fully nonlinear potential solver accurately predicts nonlinear and dispersive phenomena. It also proves that viscous effects and vorticity close to the free surface do not seem to affect its formation (elevation), in the examined non-breaking wave conditions.

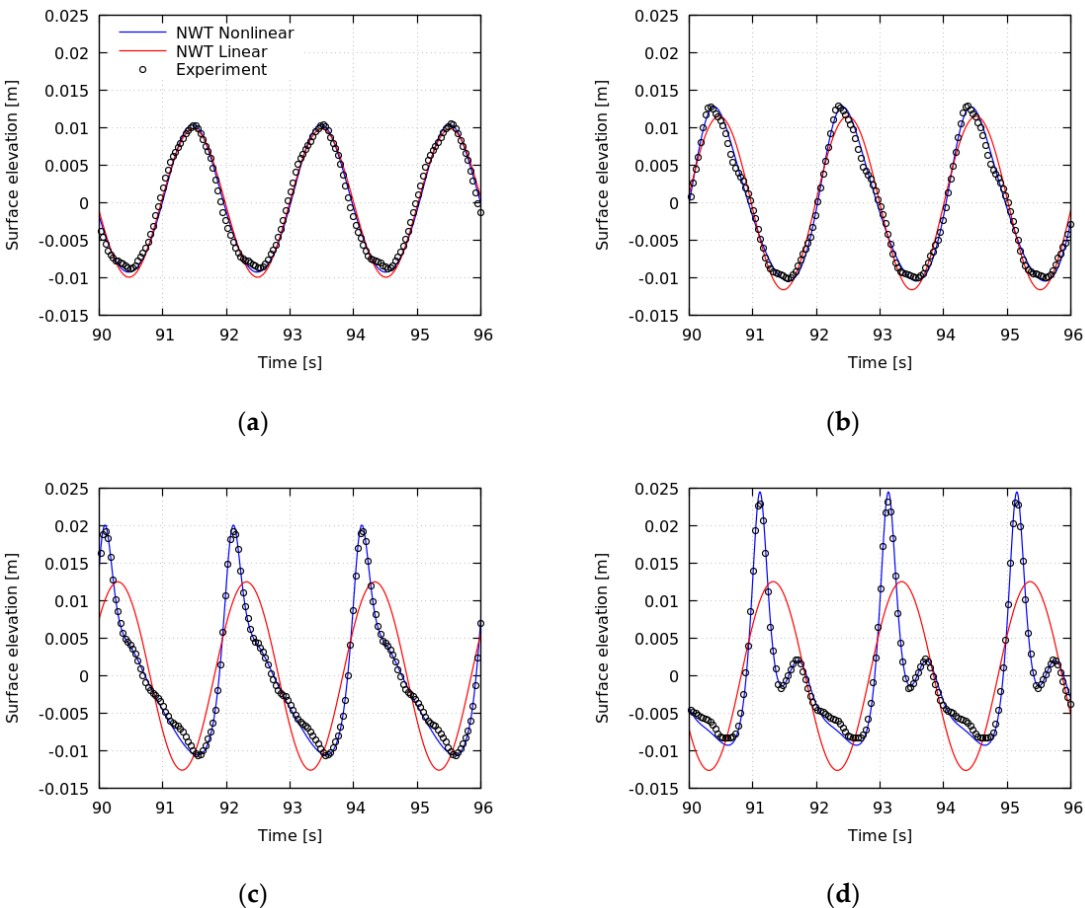

**Figure 4.** *Cont.*

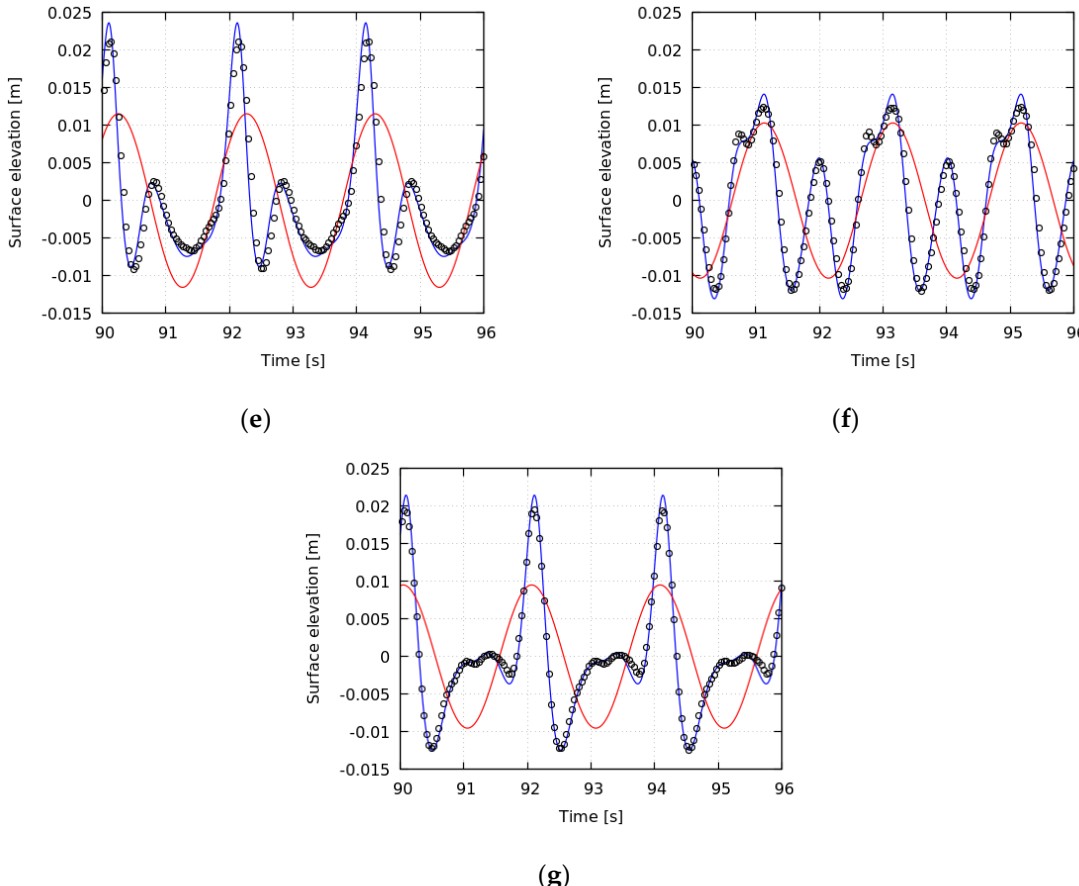

**Figure 4.** Free surface elevations at wave gauge 1–7 (see Figure 3) of a periodic wave with H = 1.905 cm and T = 2.02 s, interacting with variable bathymetry. Comparison between numerical predictions and experimental data. (**a**) Wave gauge 1 at $x$ = 5.7 m, (**b**) wave gauge 2 at $x$ = 10.5 m, (**c**) wave gauge 3 at $x$ = 12.5 m, (**d**) wave gauge 4 at $x$ = 13.5 m, (**e**) wave gauge 5 at $x$ = 14.5 m, (**f**) wave gauge 6 at $x$ = 15.7 m, and (**g**) wave gauge 7 at $x$ = 17.3 m.

*4.2. Generation and Absorption of Periodic Waves with Very High Wave Steepness*

Numerical wave generation and absorption of nonlinear periodic waves are considered in a rectangular computational domain (see Figure 1). The most challenging cases of waves with very high wave steepness are considered, corresponding to wave heights of 80% of the maximum value $H_{max}$, as estimated through Equation (25) defined in Reference [30]. Three non-dimensional wavelengths are examined, $\lambda/d$ = 1, 7, and 13, that correspond to deep, intermediate, and shallow water depths. Table 1 presents the wave properties of the considered cases. Wave parameters and numerical results are defined and presented in non-dimensional form; space is non-dimensionalized by the water depth ($d$) and acceleration by gravity ($g$), so that time is non-dimensionalized by $\sqrt{d/g}$.

$$H_{\max}/d = \frac{0.141063\,(\lambda/d) + 0.0095721\,(\lambda/d)^2 + 0.0077829\,(\lambda/d)^3}{1 + 0.078834\,(\lambda/d) + 0.0317567\,(\lambda/d)^2 + 0.0093407\,(\lambda/d)^3} \tag{25}$$

In all cases, the computational domain extends to six wavelengths ($L = 6\lambda$), the absorbing layer at the beach side to two wavelengths ($L_{AB} = 2\lambda$), and the absorbing layer at the inflow side to one wavelength ($L_{WG} = \lambda$). Initially the water is at rest, and calm conditions are imposed to the free surface (zero surface elevation and velocity potential), and the mass flux coefficient is zero. The Lagrangian approach (Equations (8) and (9)) is selected for the free surface evolution, except for the end-nodes, where the semi-Lagrangian approach (Equations (10) and (11)) is applied. In order to avoid wave

breaking due to initial transients, the ramp function is defined over 20 wave periods for $\lambda/d = 1$ and over 10 wave periods for $\lambda/d = 7, 13$. In the deep-water case, the ramp function is applied over 20 wave periods in order to avoid wave braking, which would end the simulation. It is noted that, in absolute terms, the 20 wave duration is still smaller as compared to the other cases. The mass flow control loop initiates after 30 wave periods ($t/T = 30$). The integral gain is 1, 0.2, and 0.1 for $\lambda/d = 1, 7$, and 13, respectively, while the proportional gain is two times the integral one. Tuning of the PI controller is parametrically performed through time-domain numerical simulations. Fourier series in SF theory are truncated after 21, 31, and 51 terms for $\lambda/d = 1, 7$, and 13, respectively.

**Table 1.** Wave characteristics of the considered periodic waves with very high wave steepness, based on stream function (SF) theory. Non-dimensional wave length ($\lambda/d$), wave period (T/ $\sqrt{d/g}$), wave height (H/d), current velocity (U$_0$/c), and ratio of wave height to maximum height (H/H$_{max}$).

| Case | $\lambda$/d | T/ $\sqrt{d/g}$ | H/d | U$_0$/c | H/H$_{max}$ |
|---|---|---|---|---|---|
| deep water depth | 1.000 | 2.354 | 0.113 | 0.0 | 80.00% |
| intermediate water depth | 7.000 | 7.286 | 0.523 | 0.0 | 80.00% |
| shallow water depth | 13.000 | 12.040 | 0.589 | 0.0 | 80.00% |

Table 2 contains the results of the convergence study performed in order to select the spatial and temporal discretization of the considered cases presented in Table 1. The maximum absolute relative error of the surface elevation between NWT and SF solutions is provided as a function of the number of nodes on the free surface boundary per wavelength ($N_\lambda$) and the number of time steps per wave period ($N_T$). The error is defined through Equation (26) and is calculated at the position just before the absorbing layer at $x/\lambda = 4$ over the last ten wave periods (i.e., $t/T = 140$–150).

$$e = \frac{\zeta_{NWT} - \zeta_{SF}}{A} \cdot 100 \qquad (26)$$

**Table 2.** Dependence of the maximum absolute relative error (%) of the surface elevation between numerical wave tanks (NWT) and SF solutions at $x/\lambda = 4$ over the last ten wave periods (i.e., $t/T = 140$–150) on the number of time steps per wave period $N_T$ and the number of nodes on free surface boundary per wavelength $N_\lambda$. Bold entries indicate the selected numerical parameters.

| $N_T$ | Deep Water ($\lambda$/d = 1) | | | Intermediate Water ($\lambda$/d = 7) | | | Shallow Water ($\lambda$/d = 13) | | |
|---|---|---|---|---|---|---|---|---|---|
| | $N_\lambda = 40$ | $N_\lambda = 7$ | $N_\lambda = 100$ | $N_\lambda = 40$ | $N_\lambda = 70$ | $N_\lambda = 100$ | $N_\lambda = 100$ | $N_\lambda = 130$ | $N_\lambda = 160$ |
| 20 | 5.07 | - | - | 10.92 | - | - | - | - | - |
| 40 | 2.76 | 0.73 | - | 7.07 | 2.23 | 0.88 | 6.51 | - | - |
| 60 | 2.15 | 0.45 | **0.24** | 7.10 | 1.79 | **0.86** | 4.67 | 2.16 | **0.84** |
| 80 | 2.03 | 0.39 | 0.27 | 6.18 | 1.58 | 0.79 | 4.69 | 2.13 | 0.97 |

In the above equation, $A$ is the wave amplitude and $\zeta_{NWT}$, $\zeta_{SF}$ correspond to surface elevation predictions by NWT and SF theory, respectively.

Empty cells in the table correspond to setups that led to unstable wave solutions. In order to obtain wave solutions with less than 1% maximum relative error in surface elevation, as compared to the SF solution, 100 nodes per wavelength are used in the free surface boundary for the deep and the intermediate water-depth case ($\lambda/d = 1, 7$) and 160 in the shallow water-depth case ($\lambda/d = 13$), in combination with 60 time steps per wave period (see bold entries in Table 2). In addition, in all cases, the side walls are discretized with 80 nodes and the seabed with 100 nodes per wavelength.

Figures 5 and 6 provide an overall assessment of the accuracy and consistency of the performed simulations ensuring that the global errors remain bounded and converge to steady-state values. The upper plot of Figure 5 presents time series of the free surface elevation over 150 wave periods as

predicted by the fully nonlinear NWT for the deep water case ($\lambda/d$ = 1). The "numerical wave gauge" is placed just before the absorbing layer on the beach side, four wavelengths from the inflow boundary. The plot demonstrates the robustness of the developed NWT and the efficiency of the absorbing layer, leading to stable simulations without modulation of the wave height for 150 wave periods. The effect of the ramp function is visible in the beginning of the simulation. Its role is crucial in avoiding wave breaking by the initial transient.

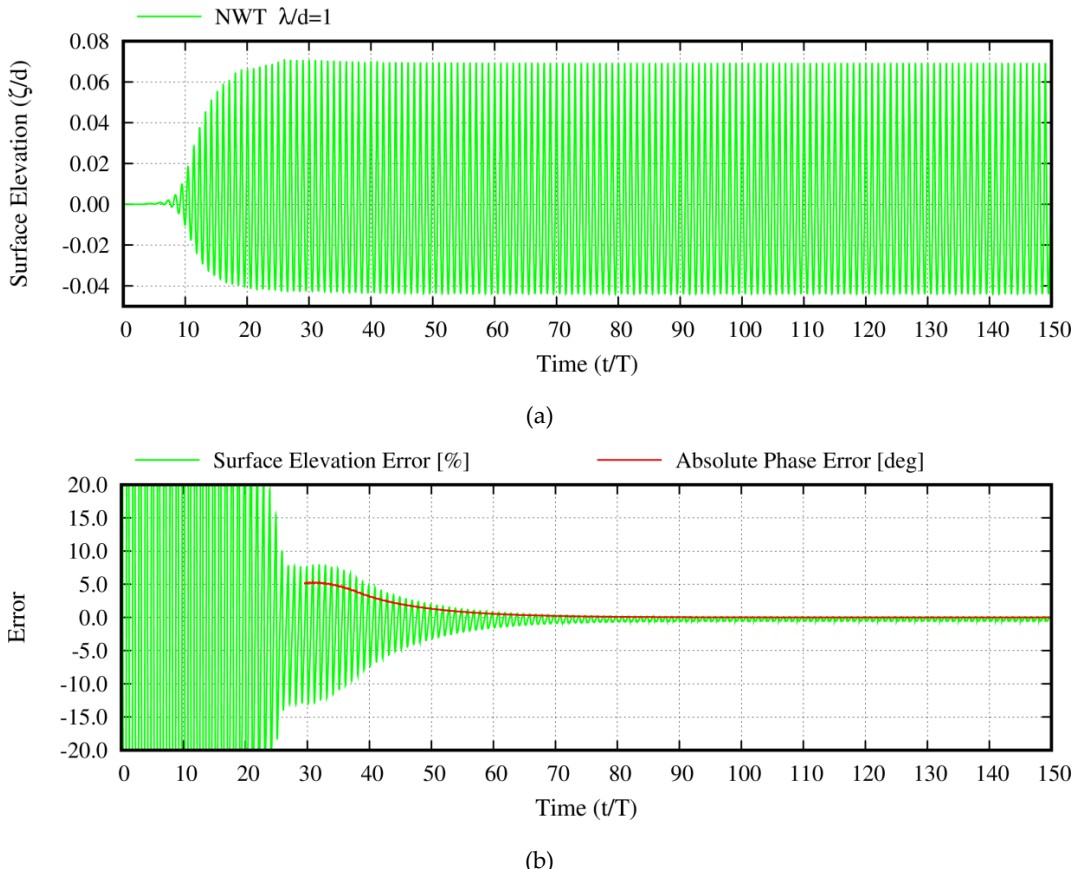

(a)

(b)

**Figure 5.** Time series over 150 wave periods of the (**a**) surface elevation predictions by NWT and SF solutions at $x/\lambda$ = 4, (**b**) surface elevation relative error and absolute phase error between NWT and SF solutions at $x/\lambda$ = 4 ($\lambda/d$ = 1, $H/H_{max}$ = 0.8, $N_\lambda$ = 100, $N_T$ = 60).

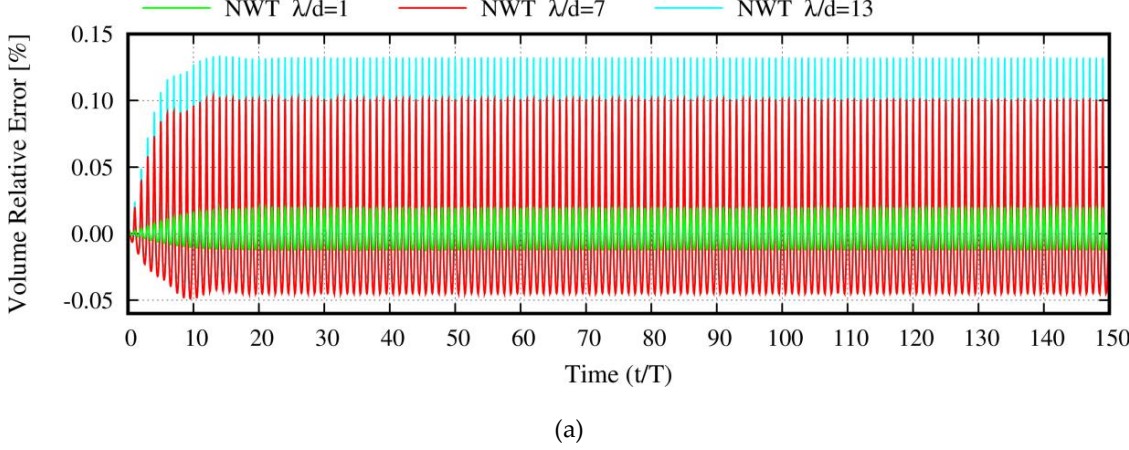

(a)

**Figure 6.** *Cont.*

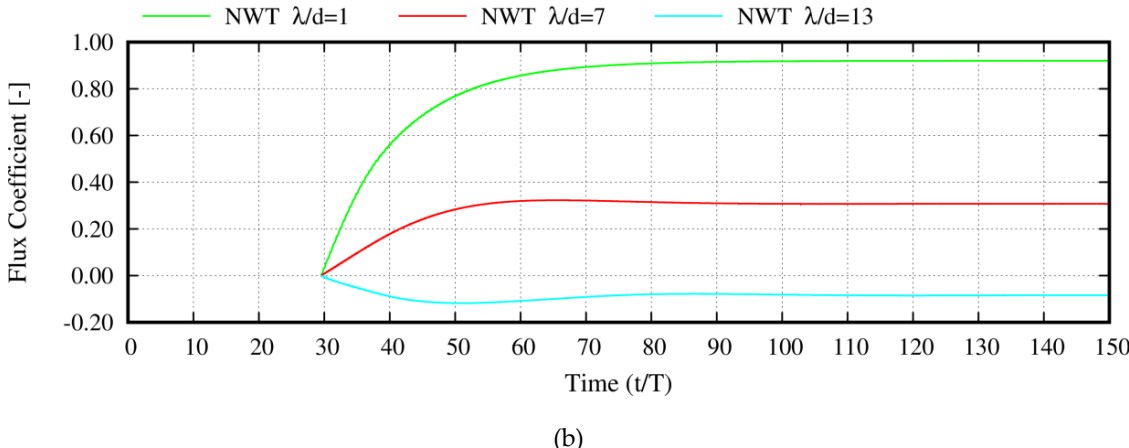

(b)

**Figure 6.** Time series over 150 wave periods of the (**a**) relative error in volume of the computational domain and (**b**) mass flux coefficient ($\lambda/d$ = 1, 7, 13, $H/H_{max}$ = 0.8).

In the lower plot of Figure 5, time series of the relative error in surface elevation and the absolute error in phase between the NWT and the SF solutions are shown for the same case ($\lambda/d$ = 1), calculated at the same position just before the absorbing layer at $x/\lambda$ = 4. Over the first 25 wave periods, the relative error in surface elevation is extremely high due to the application of the ramp function. In between $t/T$ = 25 and 30, the error attains a periodic variation with high amplitude of 10–15% due to the phase error in surface elevation, as detailed next. At $t/T$ = 30 the mass flow control loop is activated and both errors gradually start to drop, until they reach their final periodic state after about 70 periods ($t/T$ = 70), which corresponds to maximum surface elevation error 0.4% and maximum phase error 0.001°.

The upper plot of Figure 6 presents, for the three cases considered ($\lambda/d$ = 1, 7, 13), time series of the relative error in the volume of the computation domain, calculated between two successive time steps. Initially, the error oscillates due to the transient that quickly fades out. After about 10–20 wave periods, the mean value of the error in volume remains constant and close to zero for all cases, indicating convergence to a periodic state, as well as stability and consistency in the simulations.

The lower plot of Figure 6 presents time series of the flux coefficient for the three cases ($\lambda/d$ = 1, 7, 13). Initially the coefficient is zero, while after activation of the PI control loop at $t/T$ = 30, it first undergoes a relatively steep change which is followed by a smooth convergence to a fixed value. In the deep-water case ($\lambda/d$ = 1), the coefficient is close to 1, while in the intermediate ($\lambda/d$ = 7) and shallow water ($\lambda/d$ = 13) cases, the asymptotic limits are 0.31 and −0.07, respectively. According to Equation (19), when the flux coefficient is equal to 1, the mean mass flow entering the computational domain over one wave period will entirely exit from the beach side and therefore the absorbing layer does not cause a mass deficit. A positive value in the range (0, 1) indicates that part of the mean mass flow has been eliminated by the absorbing layer, while the negative value in the shallow water case indicates that additional mass enters the domain, in order to compensate the mass reduction caused by the absorbing layer, being higher than the mean mass flow. Note that, in the $\lambda/d$ = 13 case, the wave height is comparable to the water depth ($\zeta/d$ = 0.589), so the absorbing layer causes a significant mass deficit.

The upper plot of Figure 7 presents time series of the surface elevation over the last five wave periods ($t/T$ = 145–150) for the three considered water depths ($\lambda/d$ = 1, 7, 13). The agreement between the nonlinear NWT and the SF solution is excellent in all cases. This is also demonstrated in the lower plot of the same figure, in which the time series of the surface elevation relative error are presented. The maximum relative error is 0.8%, indicating that (a) the reflections are almost negligible and (b) the wave dispersion relation is accurately captured by the fully nonlinear NWT. When either of the above requirements is not fulfilled, the relative error in surface elevation will drastically increase due to either the increase in the elevation caused by reflections or the gradually increasing (toward the end-boundary) induced phase lag.

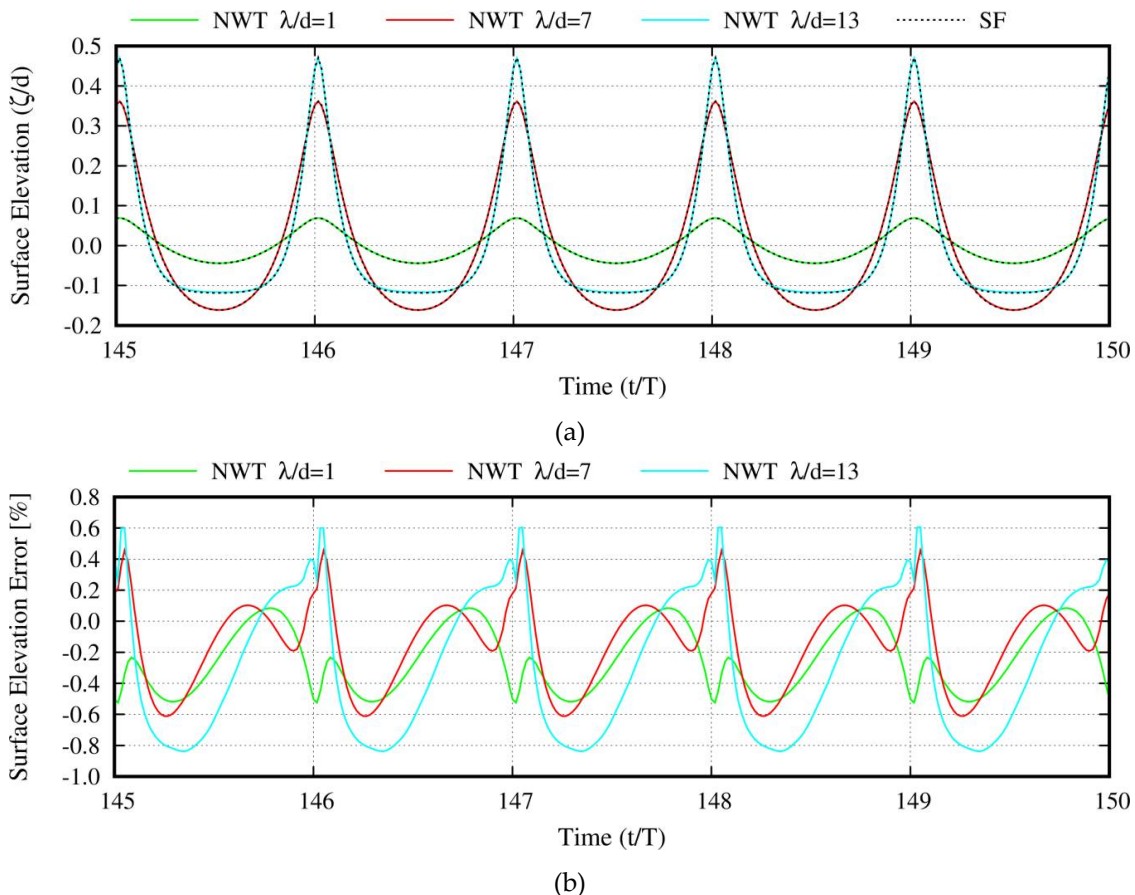

**Figure 7.** Time series over the last five wave periods of the (**a**) surface elevation predictions by NWT and SF solutions at $x/\lambda = 4$ and (**b**) surface elevation relative error between NWT and SF solutions ($\lambda/d = 1$, $H/H_{max} = 0.8$).

The three plots of Figure 8 show snapshots of the free surface elevation along the computational domain for the three simulated cases ($\lambda/d = 1, 7, 13$) after 150 wave periods ($t/T = 150$), as predicted by the fully nonlinear NWT and the imposed SF nonlinear theory. The vertical black line at $x/\lambda = 4$ indicates the starting position of the absorbing layer. The comparison between the two methods is excellent for all water-depth cases everywhere in the flow domain (with the exception of the absorbing layer), although the wave steepness is very high, being 80% of the breaking limit. In all cases, the wave crest is higher than the trough, indicating the nonlinearity of the considered waves, while in the shallow water-depth case, a "cnoidal" shape is depicted.

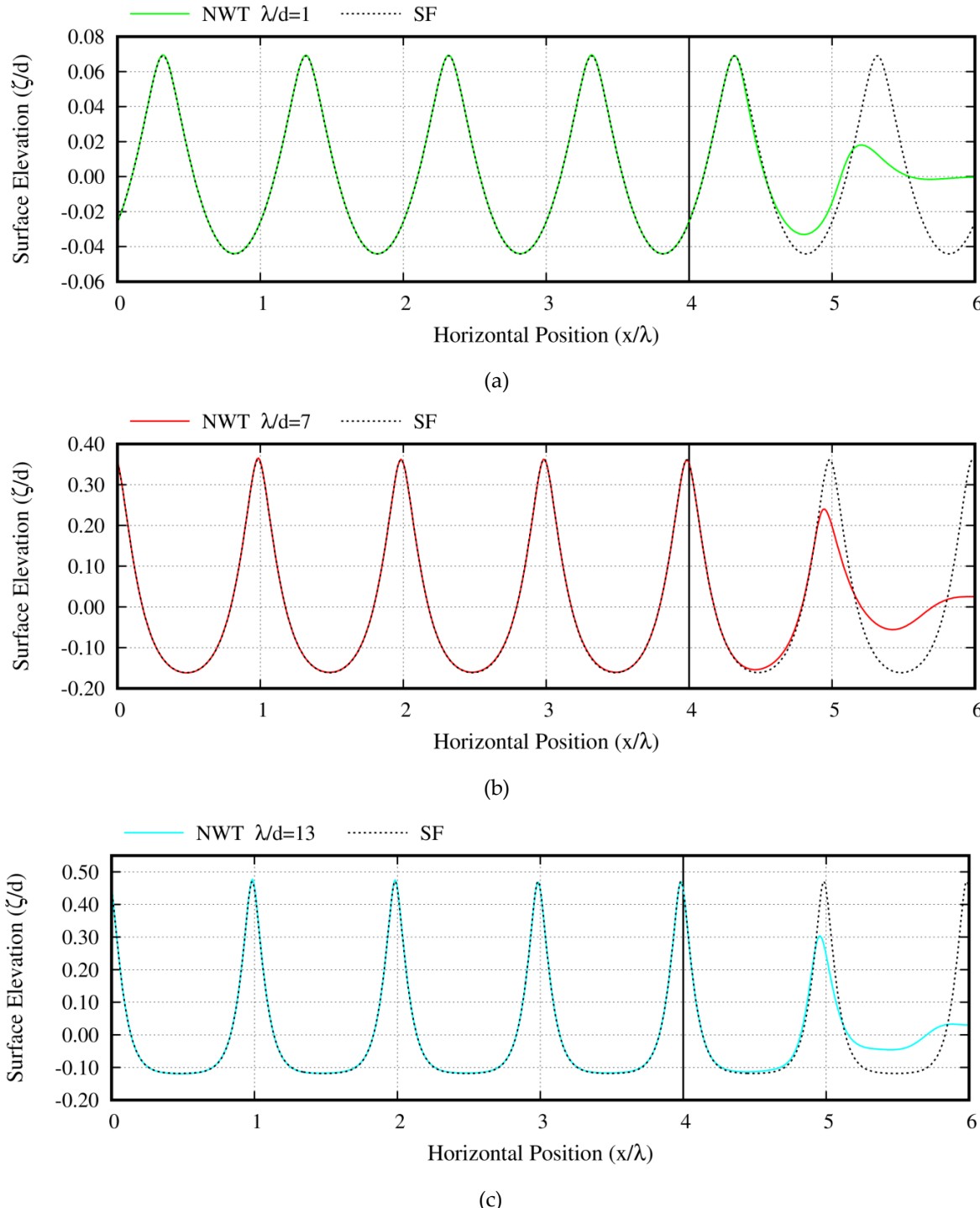

**Figure 8.** Snapshot of the surface elevation predictions by NWT and SF solutions along the computational domain at $t/T = 150$ for the three water depths: (**a**) $\lambda/d = 1$, (**b**) $\lambda/d = 7$, and (**c**) $\lambda/d = 13$ ($H/H_{max} = 0.8$). The absorbing layer starts at the vertical line ($x/\lambda = 4$).

In order to demonstrate the effect of the modified boundary condition applied to the absorbing beach side (see Equation (19)), the following three alternative modeling options were considered for the intermediate depth case ($\lambda/d = 7$): (a) the flux coefficient is estimated based on the PI control element, (b) the outgoing mass flux is set equal to the current velocity (see Equation (6)) so no additional mass flux crosses the end-wall ($c_{flux} = 0$), and (c) the outgoing mass flux is set equal to the wave mean mass transport $U_S$ ($c_{flux} = 1$). The upper plot of Figure 9 shows the snapshot of the free surface elevation

along the computational domain after 150 wave periods ($t/T = 150$), as predicted by these three alternative boundary conditions and the SF imposed solution (similar to the middle plot in Figure 8). By setting the correct mass flux, as estimated through the control loop (PI control), the wave length of the generated wave and, in turn, its dispersion relation are almost identical to the SF-imposed solution.

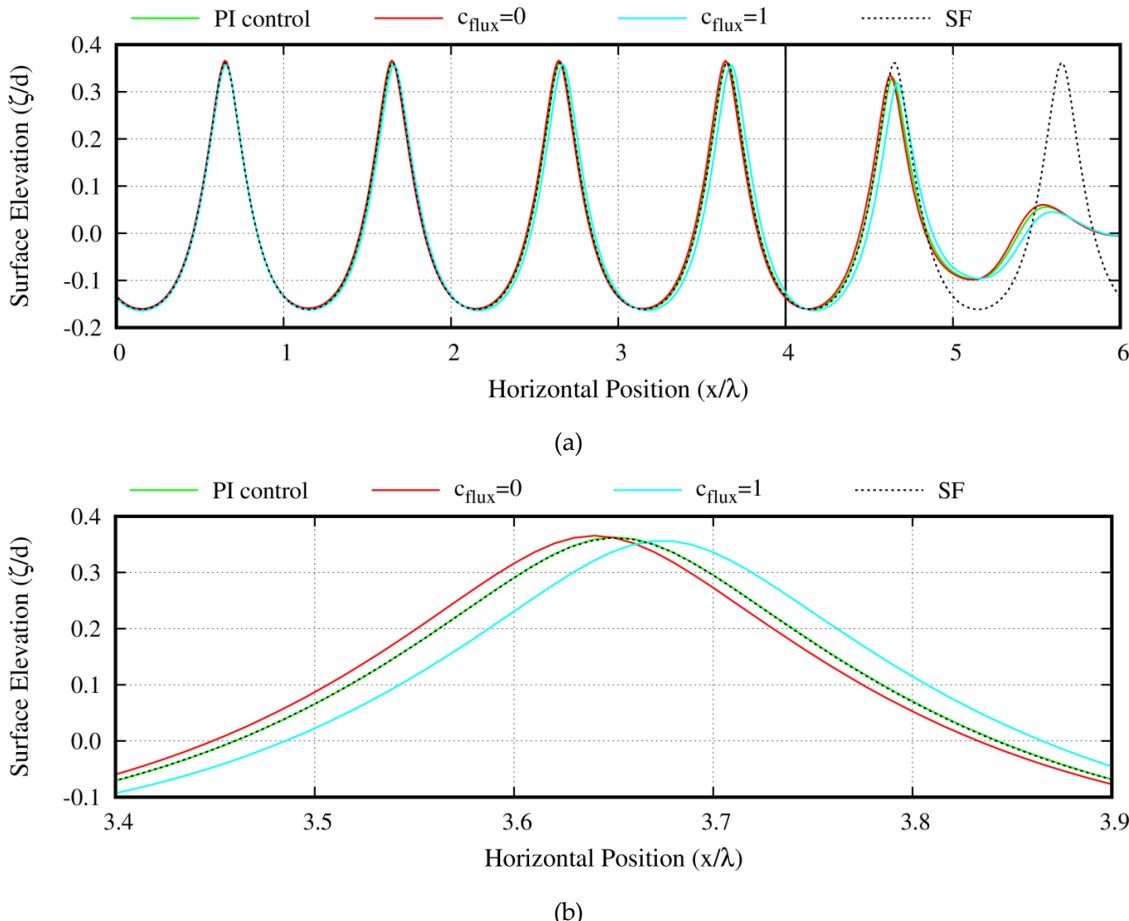

**Figure 9.** Comparison between different boundary conditions at the absorbing beach side. Snapshot of the surface elevation predictions by NWT and SF solutions at $t/T = 150$ ($\lambda/d = 7$, $H/H_{max} = 0.8$), (**a**) along the computational domain, and (**b**) close-up view of the fourth wave crest just before the absorbing layer.

On the other hand, zero mass flux ($c_{flux} = 0$) adds resistance to the flow that causes a slight reduction in wavelength (which is depicted in the plot as space lagging), in comparison to the ideal case in which periodic conditions are considered. A mass flux higher than the optimum value estimated by the controller ($c_{flux} = 1$) has the opposite effect, causing a slight increase in wavelength. This is clearly seen in the close-up in the lower plot of Figure 9 that focuses on the wave crest of the fourth wavelength just before the wave enters the absorbing layer. It is worth noting that the modified boundary condition affects the wavelength of the generated wave, but it does not reduce the efficiency of the absorbing layer. Absorbing is very efficient in all three cases.

### 4.3. Generation and Absorption of Periodic Waves Interacting with A Steady, Uniform Current

Numerical wave generation and absorption of nonlinear periodic waves interacting with a steady, uniform current are considered in a rectangular computational domain (see Figure 1). The intermediate depth case ($\lambda/d = 7$) is selected, considering a wave with relatively high wave steepness (wave height

50% of the maximum value) traveling on an opposing and a coplanar steady current of non-dimensional velocity $U_0/c = \pm 0.2$. Table 3 summarizes the wave properties of the considered cases.

**Table 3.** Wave characteristics of the considered periodic waves interacting with a steady, uniform current, based on SF theory. Non-dimensional wave length ($\lambda/d$), wave period (T$/\sqrt{d/g}$), wave height ($H/d$), current velocity ($U_0/c$), and ratio of wave height to maximum height ($H/H_{max}$).

| Case | $\lambda/d$ | $T/\sqrt{d/g}$ | $H/d$ | $U_0/c$ | $H/H_{max}$ |
|---|---|---|---|---|---|
| opposing current | 5.103 | 7.606 | 0.327 | −0.2 | 56.62% |
| no current | 7.000 | 7.606 | 0.327 | 0.0 | 50.00% |
| coplanar current | 8.678 | 7.606 | 0.327 | 0.2 | 47.32% |

The numerical setup of Section 4.2 is used for the domain size and the boundary discretization in BEM. Similar to the previous section, initially, calm conditions are imposed to the free surface (zero surface elevation and velocity potential), as well as zero current velocity and mass flux coefficient. The Lagrangian approach is selected for the free surface evolution. A smaller time step is used, equal to $T/80$, in order to ensure that the free surface particles in the Lagrangian approach will not cross the vertical boundaries due to the additional current velocity and cause numerical instabilities. Application of the ramp function extends over two wave periods, given that the breaking limit is not so close. The integral gain in the control loop is equal to 0.05, 0.2, and 0.5 for the opposing, the zero, and the coplanar current, respectively, while the proportional gain is two times the integral one. Fourier series in SF theory are truncated after 31 terms.

The upper plot of Figure 10 presents the time series of the relative error in volume of the computation domain over 150 wave periods for the three current cases considered ($U_0/c = 0, \pm 0.2$). Initially, the error oscillates due to transient phenomena that quickly fade out. After 10–15 wave periods, the mean value of the error in volume becomes constant and close to zero, justifying the stability and consistency of the wave–current interaction simulation.

The middle and the lower plot of Figure 10 present time series of the absolute phase error of the surface elevation (calculated at $x/\lambda = 4$) and of the mass flux coefficient for the three current velocities. As in the previous case of Section 4.2, initially the flux coefficient is zero, while, after activation of the PI control loop at $t/T = 30$, it undergoes a relatively steep change, which is followed by a smooth convergence to the steady-state value at $t/T = 70$–80. The converged values of the coefficient are 0.13, 0.24, and 0.47 for the opposing, the zero, and the coplanar current, respectively, indicating that the flux coefficient attains higher steady-state values as the current velocity increases in the direction of wave propagation. The phase error (input signal in the control loop) gradually decreases after activation of the PI control loop at $t/T = 30$ and becomes zero at $t/T = 70$–80, following the increase in the mass flux coefficient (control variable) that converges to its steady value at the same time instance.

In the upper plot of Figure 11, the snapshot of the free surface elevation along the computational domain after 150 wave periods ($t/T = 150$) is presented for the three current velocities. Overall, the agreement between the predictions by NWT and SF theory is excellent. As expected, the opposing current leads to steeper and shorter waves, while the coplanar current has the opposite effect. Although in general the opposing current increases the wave height, while the coplanar current decreases it, in the present study, the same (current affected) wave height was imposed in the SF solution, assuming that the interaction between the wave and the current was already accounted for. Moreover, the inherent nonlinearity of the wave–current interaction problem affects the elevation of the wave crest and trough, as depicted in the figure. A coplanar current slightly increases the elevation of the crest and trough, while an opposing current reduces the elevation values, in comparison to the zero-current case [36].

In the lower plot of Figure 11, the relative error in surface elevation is shown, estimated based on Equation (26) along the computational domain at the same time instance ($t/T = 150$) for the three current velocities. Maximum error does not exceed 0.6% (except inside the absorbing layer), indicating that the numerical generation and absorption of waves traveling on a steady, uniform current is efficiently and

accurately simulated in the developed NWT with the modified boundary condition on the beach side. The same conclusion is drawn by comparing the time series of the NWT with the SF imposed solution over the last five wave periods, which are calculated just before the absorbing layer, at $x/\lambda = 4$, and are shown in Figure 12.

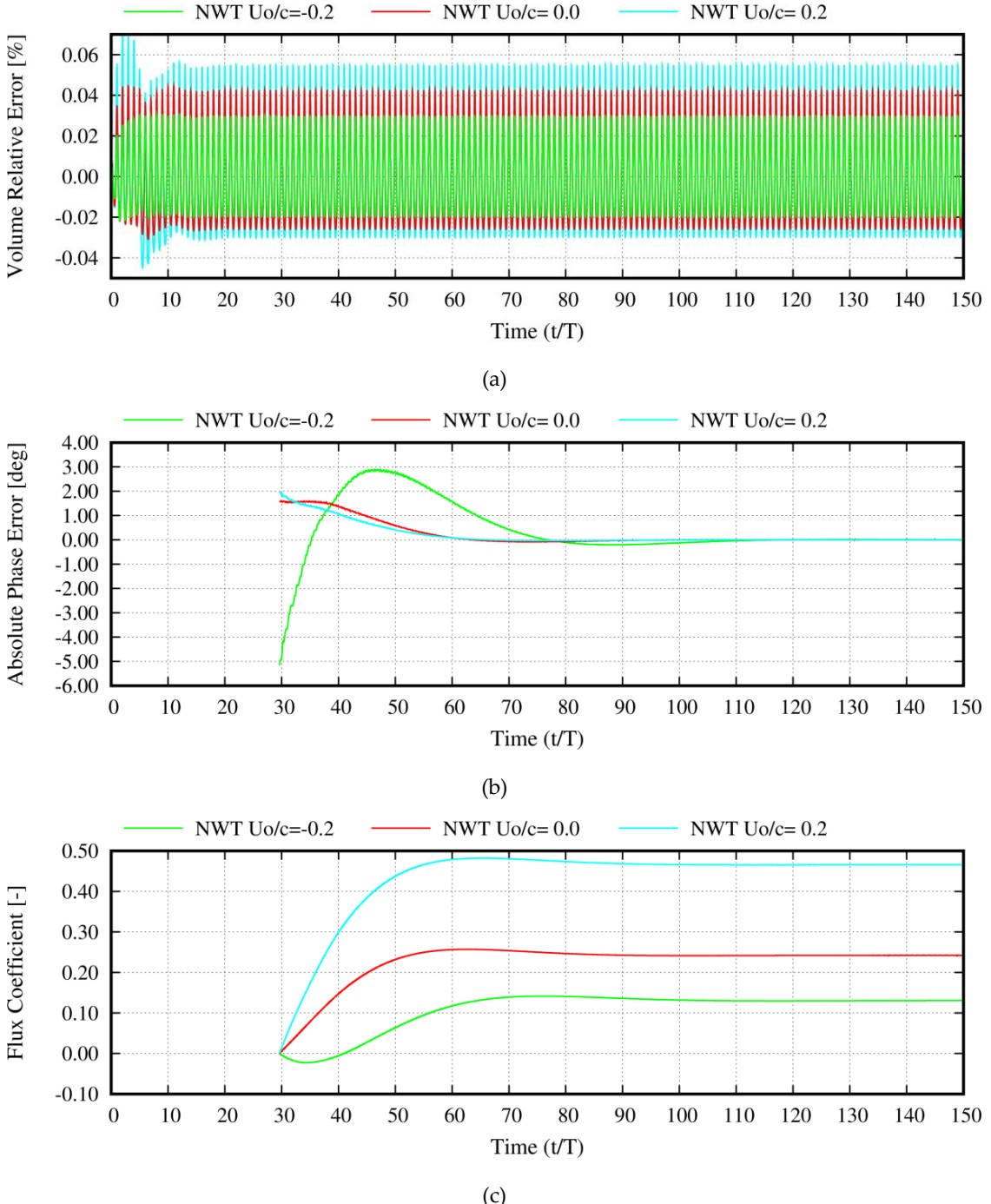

(a)

(b)

(c)

**Figure 10.** Time series over 150 wave periods of the (**a**) relative error in volume of the computational domain, (**b**) absolute phase error of the surface elevation between NWT and SF solutions, and (**c**) mass flux coefficient calculated ($U_0/c = 0, \pm0.2, \lambda/d = 7, H/H_{max} = 0.5$).

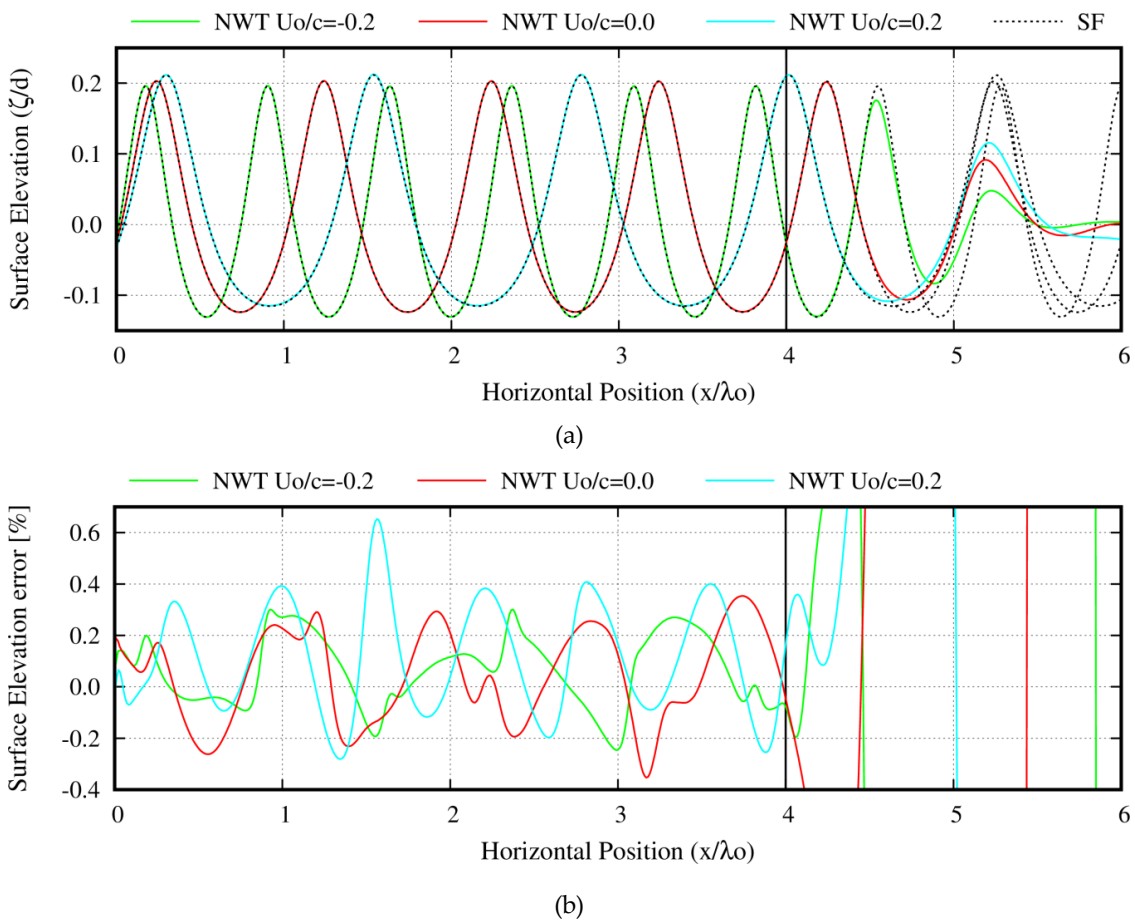

**Figure 11.** Snapshot of the (**a**) surface elevation predictions by NWT and SF solutions and (**b**) surface elevation relative error along the computational domain at $t/T = 150$ ($U_0/c = 0, \pm 0.2, \lambda/d = 7, H/H_{max} = 0.5$).

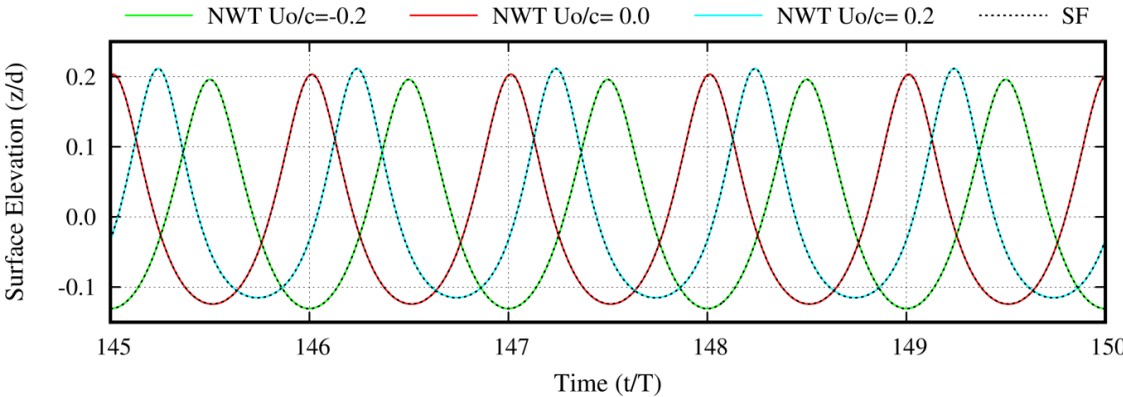

**Figure 12.** Time series over the last five wave periods of the surface elevation predictions by NWT and SF solutions at $x/\lambda = 4$ ($U_0/c = 0, \pm 0.2, \lambda/d = 7, H/H_{max} = 0.5$).

### 4.4. Sensitivity of the Modified Boundary Condition

The control loop will inherently account for numerical errors due to discretization. Consequently, flux coefficient ($c_{flux}$) depends on the numerical parameters that define the spatial and temporal discretization. In order to assess the sensitivity of the control loop to the numerical discretization, a parametric study is performed based on the configurations tested in Table 2. The converged steady-state values of the estimated flux coefficients are presented in Table 4. The simulations with 20 time steps

per wave period ($N_T$ = 20), which provide increased errors (see Table 2), are included in the two tables, to highlight the ability of the control loop to account for the discretization errors. This is reflected on the significantly different values of the flux coefficient between the configurations with poor and dense discretization (compare the errors in Table 2 with the corresponding flux coefficients in Table 4). On the other hand, the flux coefficient is found to converge rather rapidly to its steady-state value, so its dependence on the discretization is rather insignificant for valid grid-independent wave solutions. It is noted that the phase error (control variable) in all considered configurations successfully converges to 0°.

**Table 4.** Steady-state value of the flux coefficient versus the number of time steps per wave period ($N_T$) and the number of nodes on free surface boundary per wavelength ($N_\lambda$). Bold entries indicate the selected numerical parameters.

| $N_T$ | Deep Water ($\lambda/d$ = 1) | | | Intermediate Water ($\lambda/d$ = 7) | | | Shallow Water ($\lambda/d$ = 13) | | |
|---|---|---|---|---|---|---|---|---|---|
| | $N_\lambda$ = 40 | $N_\lambda$ = 70 | $N_\lambda$ = 100 | $N_\lambda$ = 40 | $N_\lambda$ = 70 | $N_\lambda$ = 100 | $N_\lambda$ = 100 | $N_\lambda$ = 130 | $N_\lambda$ = 160 |
| **20** | 1.67 | - | - | 0.78 | - | - | - | - | - |
| **40** | 1.04 | 1.05 | - | 0.40 | 0.38 | 0.33 | 0.30 | - | - |
| **60** | 0.89 | 0.97 | **0.95** | 0.32 | 0.33 | **0.31** | 0.15 | 0.02 | **−0.07** |
| **80** | 0.85 | 0.94 | 0.94 | 0.27 | 0.30 | 0.31 | 0.10 | −0.01 | −0.08 |

In addition, as mentioned in Section 3.6, dispersion errors are attributed to the effect the absorbing layers have on the wave solution, as compared to the standard periodic formulation with periodic boundary conditions applied on the lateral boundaries. They are generated due to the interaction of the wave solution with the absorbing layer. Thus, for valid grid-independent wave solutions at which the effect of discretization is minor, dispersion errors (and flux coefficient) mainly depend on the parameters that define the wave and the absorbing layers.

As far as the wave parameters are concerned, the flux coefficient depends on (a) the non-dimensional wavelength ($\lambda/d$), as depicted in the lower plot of Figure 6 or in Table 4; (b) the wave steepness ($H/\lambda$), as derived by comparing the coefficient values between wave heights of 50% (0.24) and 80% (0.31) of the maximum value in the intermediate water-depth case presented in the lower plots of Figures 6 and 10, respectively; and (c) the Froude number ($Fr = U_0 / \sqrt{gd}$), as shown in the lower plot of Figure 10.

Regarding the absorbing layer, the flux coefficient depends on its length and on the damping terms ($damp_i$ in Equations (16) and (17)). The latter include the choice of the damping function $v_i(x)$, the considered variable (i.e., the potential or its normal derivative), and the equation in which dissipation is added. However, it is noted that this is not supported by the presented numerical results, since the parameters of the absorbing layer remain unaffected.

## 5. Conclusions

A fully nonlinear potential NWT was formulated based on the Boundary Element Method, with plane elements carrying linearly distributed singularities, and on the mixed Eulerian–Lagrangian formulation of the free surface equations. In the first part of the paper, emphasis was given to the important implementation details needed to consistently and accurately perform numerical wave generation and absorption of periodic waves. The key points are summarized next.

In connection to the BEM solver, (a) the integrals are analytically evaluated for improved accuracy, (b) the double node implementation assures unique definition of the velocity at the corners of the computational domain preventing the so-called saw-tooth instability, and (c) regridding of the free surface panels (in the Lagrangian formulation) increases the accuracy in BEM and maintains an almost constant panel resolution that is necessary in the wave–current interaction case.

In connection to the Lagrangian part, (a) a fourth-order Runge–Kutta time integration scheme is applied, and (b) both end-boundaries remain fixed in the x direction by consistently tracking the surface end-nodes only in the vertical direction based on the semi-Lagrangian approach.

Regarding the inflow conditions, a stream function numerically exact nonlinear wave solution is imposed to the inflow boundary, which inherently accounts for the nonlinear wave–current interaction of period waves with a steady, uniform current.

Regarding the far-field conditions, (a) absorbing layers are introduced on both sides of the computational domain, extending to one and two wavelengths at inflow and outflow boundaries, respectively; (b) damping terms are added in both free surface evolution equations; (c) the damping term in Bernoulli equation is proportional to $\partial_n\varphi$; and (d) the boundary condition on the vertical end-boundary is modified to consider an additional unknown mass flux. Correction of the mass flux through the domain, by means of a PI controller, accurately satisfies the dispersion relation (and wavelength) of the generated waves, as compared to the standard periodic formulation without absorbing layers.

The fully nonlinear NWT was firstly validated against measurements in the interaction of periodic waves with variable bathymetry case. The comparison verified that the solver accurately predicts nonlinear and dispersive phenomena.

Next, accurate nonlinear periodic wave solutions (not necessarily restricted by the zero mass flux requirement) were obtained in the fully nonlinear NWT, for wave heights up to 80% of the maximum value over 150 wave periods (time instant at which the numerical simulation is forced to stop by the user). The SF wave solution was matched at the inflow boundary, and valid wave solutions were produced for shallow, intermediate, and deep water depths, also considering the contribution of a steady, uniform current. Absorbing layers rendered wave absorption at the beach side very efficient (at the highest wave amplitude, the maximum surface elevation relative difference between fully NWT and SF solution is less than 0.8%).

This was accomplished by thoroughly working out the numerical implementation details. Specifically, the modified boundary condition on the vertical end-boundary compensates the dispersion error induced by the absorbing layers on the wave solution, as compared to the standard periodic formulation with periodic boundary conditions applied on the lateral boundaries. It is introduced to account for the mean mass transport appearing in nonlinear waves due to Stokes' drift and for the mass reduction caused by the absorbing layers.

Finally, the flux coefficient, the free parameter that defines the additional mass flux over the end-boundary, is found to converge rather rapidly to its steady-state value, so its dependence on the discretization is rather insignificant for valid grid-independent wave solutions. It mainly depends on non-dimensional wavelength ($\lambda/d$), wave steepness ($H/\lambda$), and Froude number ($Fr = U_0/\sqrt{gd}$), as well as on the parameters that define the absorbing layers.

**Author Contributions:** Methodology, D.I.M. and S.G.V.; software, D.I.M.; supervision, V.A.R. and S.G.V.; writing—original draft, D.I.M.; writing—review and editing, V.A.R. and S.G.V. All authors have read and agreed to the published version of the manuscript.

**Funding:** This research received no external funding.

**Conflicts of Interest:** The authors declare no conflict of interest.

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
