# Peer review of "Generation and Absorption of Periodic Waves Traveling on a Uniform Current in a Fully Nonlinear BEM-based Numerical Wave Tank"

_jmse, doi:10.3390/jmse8090727_

Round 1

Reviewer 1 Report

  • Abstract/introduction: Would like to see a stronger motivation for why this work has been performed, especially towards the best practices of NWTs in the industry today.
  • Line 102: Please ellaborate a bit on why you want to cancel out the 2nd order drift when it is real.
  • Line 150: Did it diverge after 150 periods or did you stop the simulation?
  • Chapter 4: Would like to include a picture of one of the mesh configurations used.
  • Figure 3: Which of the configurations in Table 2 is this?
  • Results in Chapter 4: Would like to see a comparison between present results, experiments and traditional potential flow codes like Wamit or Aqwa that also have higher order waves implemented. I.e. to better understand the benefits of this work (see also comment about Introduction). However, I understand that these are data that may not be available to you.

Reviewer 2 Report

Please, see the file.

Reviewer 3 Report

The authors developed a two-dimensional, BEM-based numerical wave tank for discussing wave generation and absorption. In fact, this study is far from novel research, since the NWT has been well-developed for decades, refer to ref. [3]. I can realize that it is not easy to do something new on this topic. In my opinion, the main contribution of the authors is to modify the boundary condition for the numerical beach. The structure of this paper is fine, the description of the numerical model is clear, and the result is reasonable. Thus, this study may be helpful for readers who want to build their own NWT. However, there still have some descriptions need to be clarified.

  1. Line 166, the description of the non-penetration condition on the wall boundary (SAB) seems not appropriate for the steady current.
  2. Lines 197 and 202, is Φ should be φ?
  3. In Eq(13), Line 212 and 215, is φs should be φ?
  4. Why the authors use the dimensionless parameters in Tables 1 & 4, and what the physical meaning these parameters were?
  5. Lines 355-356, why the authors use 20 wave periods in the ramp function for the case of a deepwater wave, but that for the intermediate and the shallow water waves are only 10 wave periods.
  6. Line 357 and 471, how did the authors do to give the value of integral gain?
  7. Lines 437-440, the results without PI control seem to result in the phase difference rather than the wavelength change.
  8. Lines 515-519, this description was not proved by the author's numerical results. Thus, it is not appropriate to write in the conclusion.
